# Early Effects of Nivolumab and Ipilimumab Combined Immunotherapy in the Treatment of Metastatic Melanoma in Poland: A Multicenter Experience

**DOI:** 10.3390/biomedicines10102528

**Published:** 2022-10-10

**Authors:** Renata Pacholczak-Madej, Aleksandra Grela-Wojewoda, Mirosława Puskulluoglu, Joanna Lompart, Manuela Las-Jankowska, Katarzyna Krawczak, Ewa Wrona, Lech Zaręba, Justyna Żubrowska, Jerzy Walocha, Stanisława Bazan-Socha, Marek Ziobro

**Affiliations:** 1The Maria Skłodowska-Curie National Research Institute of Oncology, Kraków Branch, 31-115 Krakó, Poland; 2Department of Anatomy, Jagiellonian University Medical College, 33-332 Kraków, Poland; 3Surgical Oncology, Ludwik Rydygier Collegium Medicum in Bydgoszcz, Nicolaus Copernicus University in Torun, Oncology Centre, 85-067 Bydgoszcz, Poland; 4Department of Clinical Oncology, Oncology Center—Prof Franciszek Lukaszczyk Memorial Hospital, 85-796 Bydgoszcz, Poland; 5Frederic Chopin Provincial Hospital No. 1, 35-055 Rzeszów, Poland; 6Department of Clinical and Laboratory Genetics, Medical University of Lodz, 92-215 Łódź, Poland; 7Department of Chemotherapy, Medical University of Lodz, Copernicus Memorial Hospital, 90-419 Łódź, Poland; 8Interdisciplinary Centre for Computational Modelling, College of Natural Sciences, University of Rzeszów, 35-310 Rzeszów, Poland; 9Holy Cross Cancer Centre, 25-734 Kielce, Poland; 10Department of Internal Medicine, Jagiellonian University Medical College, 31-066 Kraków, Poland

**Keywords:** melanoma, immunotherapy, real-life data, nivolumab and ipilimumab, adverse events

## Abstract

Nivolumab and ipilimumab combination became the first-line standard in advanced melanoma. We assessed its efficacy in a real-life study in Poland. In a one-year follow-up, we evaluated the medical records of 50 melanoma patients treated with that modality in five oncology centers. We recorded therapy outcomes and adverse events (AEs) after 3 and 12 months of therapy. At the first checkpoint, the disease control rate (DCR) was recorded in 58% (n = 29) of patients, but the same number of patients (n = 29, 58%) stopped immunotherapy due to disease progression (PD, n = 14, 48.3%), toxicity (n = 11, 37.9%) or death (n = 4, 13.8%). Among patients with DCR after the induction phase, 8 (27.6%) terminated due to toxicity, and 21 (72.4%) continued. However, at the 12-month checkpoint, only 14 patients (27% of all) were still receiving immunotherapy. In 7 (33.3%) it was discontinued due to PD (n = 2), toxicity (n = 2, 28.6% each), or death (n = 3, 42.9%). AEs occurred in 66.7% (n = 34) of patients; severe (grade 3 or 4) in half of them. Interestingly, those with AEs had an 80% lower risk of death (hazard ratio [HR] 0.2, 95% confidence interval [CI] 0.07–0.57, *p* = 0.001) and PD (HR 0.2, 95%CI 0.09–0.47, *p* < 0.0001). In the entire group of patients, after a 12-month follow-up, the median overall survival was not reached (NR, range: 6.8 months-NR) and progression-free survival was 6.3 (range: 3-NR) months. Our results demonstrate that combined immunotherapy is less effective in real-life than in pivotal trials. However, early responders will likely continue the therapy after a one-year follow-up. AEs occurrence might be a predictor of clinical effectiveness.

## 1. Introduction

Melanoma constitutes 2% of malignancies in Poland. However, in young men between 20 and 44 years, its frequency increases to 6% and is responsible for 5% of deaths. Furthermore, its prevalence in the general population has constantly risen over the past 15 years, and in 2019 in Poland, it accounted for 3800 new cases and 1400 deaths with an annual incidence of 6/100,000 [1,2]. Interestingly, its occurrence in middle Europe countries is similar to that observed in Mediterranean countries (3–5/100,000), being lower than in the Nordic region (12–34/100,000) or Australia or New Zealand (50/100,000) [3]. In 80% of cases, melanoma is diagnosed early and can be treated solely by a surgical approach. On the other hand, advanced melanoma, which constitutes 5% of the entity, remains an area of therapeutic challenge with a 5-year survival rate of around 20–40% despite proper treatment [2]. Therefore, it implied the research for new treatment modalities over the past decades.

Dacarbazine is the only chemotherapeutic agent registered in Poland and Europe to treat advanced melanoma [4]. However, its efficacy is limited, with 15% of the objective response rate (ORR) and 4 months of median response duration [2]. A unique feature of melanoma compared to other solid tumors is the high mutation burden caused by sun exposure. These mutations refer to neoepitopes forming, serving as neoantigens [5]. Therefore, high tumor immunogenicity implied research for new molecules used in immunotherapy [6].

The first agent from this group was high-dose interferon alpha-2b registered in 1996 in the adjuvant setting for high-risk melanoma. Importantly, most of the patients required dose modification due to severe toxicities [7]. Afterwards, interleukin 2 (IL-2) was approved by US Food and Drug Administration (FDA) in 1998. However, its use was related to only 16% of ORR and 12 months of median overall survival (OS) [8] with a high risk of septic shock and 2.2% of treatment-related deaths. Furthermore, among patients treated with IL-2, 44% reported grade (G) 3 hypotension, around 30% of G3 diarrhea, vomiting, and oliguria, according to the common toxicity criteria of the National Cancer Institute [9]. The subsequent adjuvant treatment affecting the immune system in melanoma was PEG-interferon alpha-2b, approved based on the EORTC 18991 clinical trial. That study included patients with stage III melanoma after resection. Unfortunately, the medication was effective as little as in ulcerated melanoma, without benefitting the whole patient population. Notably, 37% of patients discontinued treatment due to its toxicities [10]. However, despite the modest efficacy and high toxicity, these agents laid the foundation for modern oncologic immunotherapy. 

Immune checkpoint inhibitors (ICI) directed against cytotoxic T-lymphocyte antigen (CTLA)-4 or programmed cell death protein (PD)-1 and programmed death-ligand 1 (PD-L1) are essential milestones in cancer immunotherapy. Among them, the first was ipilimumab (anti-CTLA-4 antibody), registered by FDA in 2011 for advanced melanoma therapy [11]. In 2014, the FDA and one year later, the European Medical Agency (EMA) approved pembrolizumab and nivolumab, both anti-PD-1 blocking antibodies, to treat advanced/unresectable melanoma in a second line, after the failure of ipilimumab or BRAF inhibitors (in case of BRAF mutation). The Checkmate 037 study comparing nivolumab with the investigator’s choice chemotherapy (dacarbazine or carboplatin plus paclitaxel) demonstrated increased ORR (27% vs. 10%) and response duration (32 vs. 13 months) for that first option, although OS was similar in both subgroups likely to crossover [12]. Similarly, pembrolizumab improved ORR, progression-free survival (PFS), and OS in ipilimumab-refractory patients despite a 55% of crossover rate in the Keynote-002 study [13]. However, special attention deserves the Checkmate 067 study. Based on its excellent results, the FDA and EMA approved in 2015 a combined regimen of nivolumab and ipilimumab as the first-line treatment option for advanced melanoma. After 6.5 months of observation, the median OS (primary endpoint) for the combination was 72.1 months versus 36.9 months and 19.9 months for nivolumab and ipilimumab alone, however with a cost of more frequent G3/G4 adverse events (59% vs. 24% vs. 28% of patients on nivolumab + ipilimumab, nivolumab, or ipilimumab, respectively) according to Common Terminology Criteria for Adverse Events (CTCAE) [14]. Therefore, Polish and European guidelines recommend that combination as a preferred first-line treatment in advanced melanoma regardless of BRAF mutation status and especially in patients previously exposed to immunotherapy in the adjuvant setting [2,3,15]. 

Modern anticancer therapies are available in Poland in clinical trials or in National Drug Programs, which are precisely monitored via a dedicated online system [16]. The Ministry of Health reimburses agents included in these programs for those who meet the inclusion criteria. Nivolumab and pembrolizumab in monotherapy of advanced melanoma are available for Polish patients since 2016 and combined immunotherapy since October 2020 [17]. 

In this study, we evaluated the one-year efficacy of nivolumab and ipilimumab combined immunotherapy in patients with advanced melanoma treated according to the Polish National Drug Program. Data were collected from five comprehensive cancer centers in Poland and compared with results reported in clinical trials or other real-world observations. Our report might be valuable for the melanoma-oriented community since that treatment is broadly available for a short time and the real-life data are still limited. 

## 2. Materials and Methods

We retrospectively analyzed medical records from 51 patients with advanced melanoma who received combined immunotherapy from October 2020 until May 2022. Data were collected from five oncology centers specialized in melanoma treatment. We evaluated therapy efficiency and adverse events (AEs) after four months of induction therapy and a 12-month follow-up.

Inclusion criteria for combined immunotherapy in Poland are similar to those available in the Checkmate 067 study [18]. They include adult skin or mucosal melanoma patients with unresectable disease stage III or IV, without a previous regimen for advanced melanoma. Disease advancement is established based on Response Evaluation Criteria in Solid Tumors 1.1 (RECIST), according to the good performance status of the Eastern Cooperative Oncology Group (ECOG), which scored 0–1 points. Furthermore, laboratory test results must agree with summary product characteristics [19,20] and the BRAF mutation status needs to be determined. Moreover, patients with brain metastases had to be asymptomatic or after local treatment (surgery or radiation therapy). Pregnancy and breastfeeding are exclusion criteria. Initially, assessing PD-1 expression was mandatory (with an entry cut-off of >5%), but this criterion was withdrawn after the update of the Checkmate 067 study [21]. 

In our patients, treatment was administered in standard doses of ipilimumab 3 mg/kg and nivolumab 1 mg/kg applied every 3 weeks as 4 cycles of the induction therapy, followed by nivolumab flat doses of 240 mg every 2 weeks or 480 mg every 4 weeks until the disease progression (PD), unacceptable toxicity, or the patient’s consent withdrawal. 

The patient’s response to treatment was evaluated in computer tomography (CT) scans of adequate regions according to RECIST 1.1 or ‘immune’ RECIST (iRECIST) criteria. The evaluation time followed the National Drug Program criteria. The first assessment was performed between 11 and 13 weeks of the treatment and subsequent every 3–4 months or when PD was clinically suspected. In the case of pseudoprogression (defined according to iRECST as an increase in the size of lesions or the presence of new ones absent at baseline) [22], the following CT scan was recommended 4 weeks later, and the final assessment was recorded. 

We analyzed the treatment efficacy by calculating the OS (defined as the time from the start of the combined immunotherapy to patient’s death), PFS (defined as the time from the beginning of the combined immunotherapy to disease progression or patient’s death), time-to-treatment failure (TTF- defined as the time from the start of the treatment to premature discontinuation), ORR (complete remission [CR] or partial response [PR] according to RECIST 1.1 criteria in a CT scan performed after 4 series of combined immunotherapy) and DCR (CR, PR and stable disease [SD] according to RECIST 1.1 criteria in a CT scan performed after 4 series of combined immunotherapy). We also recorded AEs that occurred from the start of combined immunotherapy until the end of the observation period (May 2022), such as: skin AEs (rash, pruritus, vitiligo), endocrinopathies (thyroid gland disorders, hypophysitis, type 1 diabetes mellitus), immune-related (ir)-hepatotoxicity, colitis/diarrhea, ir-pneumonitis, rheumatological AEs, renal AEs, hematological AEs. The severity of AEs was graded using CTCAE version 5.0 (U.S. Department of Health and Human Services) [23] and they were diagnosed and managed according to the European Society for Medical Oncology guidelines [24].

We also recorded internal medicine comorbidities, including arterial hypertension (history of blood pressure > 140/90 mmHg or current antihypertensive treatment), hypercholesterolemia (a total serum cholesterol > 5.2 mmol/L or ongoing antihypercholesterolemic therapy), renal insufficiency (glomerular filtration rate [GFR] lower than 90 but higher than 60 mL/min/1.73 m^2^ [G1 in CTCAE 5.0]), heart failure (ejection fraction < 50%), ischemic heart disease (symptoms of chest pain or myocardial infarct in the past), autoimmunologic diseases (recorded in patients’ history), hypothyroidism (current use of hormone replacement therapy), type 2 diabetes mellitus (use of insulin or oral hypoglycemic agents, or a fasting serum glucose > 7.0 mmol/L) [23]. 

The study was approved by the Bioethics Committee of the National Cancer Institute in Warsaw, Poland (number 38/2022), and all patients signed appropriate consent before enrollment. 

Statistica Software version 13 and PS Imago Pro 8 (SPSS) Predictive Solutions (available from IT Services Centre of Jagiellonian University) were used in the analysis. The continuous variables were all non-normally distributed according to the Shapiro-Wilk test. They were presented as median and interquartile ranges and compared using the Mann-Whitney U-test. Categorical variables were reported as percentages and compared by χ2 test. We used the Kaplan-Meier method to estimate and visualize treatment outcomes. Cox proportional hazard regressions were used to assess the association with baseline prognostic factors. Additionally, single and multiple Cox regressions were performed to evaluate the factors influencing PFS and OS, and a log-rank test was applied to compare the survival distribution between the chosen factors. *p*-values < 0.05 were considered statistically significant. The power of Cox regression was 82% and general power for all tests was 88%.

## 3. Results

### 3.1. Baseline Patient Characteristics

The patient demographics, medical history, and baseline melanoma characteristics are shown in Table 1. Most of the patients (n = 47, 94%) presented with cutaneous melanoma, two (4%) had a mucosal subtype, and in 1 case (2%), we were not able to set the primary origin. Almost two-thirds of the patients (n = 36, 72%) were previously treated radically, and the remaining 14 individuals (28%) were primary metastatic at diagnosis. Among the former patients, 42% (n = 15) received adjuvant treatment, such as immunotherapy (nivolumab or pembrolizumab, n = 10), BRAF/MEK inhibitors (n = 3), radiotherapy (n = 1), or interferon (n = 1). The median time from the primary diagnosis to initiation of immunotherapy was one (range: 0–5) year. 

The most common internal medicine comorbidity was hypertension, diagnosed in one-third of patients, followed by hypercholesterolemia (n = 9, 18%) and diabetes mellitus type 2 (n = 5, 10%) (Table 1). In addition, one patient had heart failure, and another had renal insufficiency. In two patients, we diagnosed autoimmune diseases, namely psoriasis and rheumatoid arthritis. Of note, all comorbidities were adequately treated, well-controlled at enrollment and were not a contraindication for immunotherapy.

The basic laboratory parameters at baseline are presented in Table 1. As shown, half patients had increased lactate dehydrogenase (n = 24, 48%), the remaining parameters were within normal limits.

### 3.2. About 60% of Patients Recorded a Disease Control Rate after the Immunotherapy Induction Phase

In Figure 1, we present the chart flow, depicting the number of patients who continued or terminated treatment at 3-month and 12-month checkpoints. We also reported factors related to therapy discontinuation and subsequent treatment modalities. As shown, we qualified 51 patients for combined immunotherapy (Figure 1). One patient (2%) was early lost of follow-up and excluded from the study. 

In the first tumor evaluation, performed three months after the immunotherapy began, DCR was recorded in 58% (n = 29) of patients, with ORR documented in 17 (34%). The remaining 20 (40%) patients noted disease progression (n = 13, 26%) or died (n = 7, 14%), and one (2%) was not assessed in CT scan (Figure 1). It is noteworthy that 19 (38%) patients received less than 4 series of treatment induction mainly due to toxicity (n = 9, 47%), PD (n = 5, 26%), death (n = 4, 21%) or other medical condition (n = 1, 5%) (Figure 1). Altogether at this evaluation point, treatment was terminated in 29 individuals (58%) due to PD (n = 14, 48%) or toxicity (n = 11, 38%), or deaths (n = 4, 21%). However, in those who had a treatment withdrawal because of immunotherapy AEs, CT scan revealed DCR in 8 (73%) subjects (ORR: n = 6, 55%; SD: n = 2, 18%). Furthermore, the presence of immunotherapy AEs was the only factor influencing DCR at the early evaluation (n = 33 vs. n = 17, *p* < 0.0001). 

On the other hand, radiotherapy, parallel to systemic treatment, was applied in 12 individuals (24%) of whom 5 patients had central nervous system metastases and had no impact on ORR (*p* = 0.62) or DCR (*p* = 0.68). Similarly, other factors, including performance status, BRAF mutation, metastasis number and sites, and previous immunotherapy in an adjuvant setting, were not related to better outcomes.

Among 29 (57%) patients who demonstrated DCR in the first CT evaluation, 8 had early withdrawal from treatment due to severe toxicity, and 21 (42%) continued treatment.

### 3.3. Only 29% of Patients Continued Immunotherapy Till the 12-Month Checkpoint

At the 12-month checkpoint, only 14 (29% of all) patients continued immunotherapy (Figure 1). On the contrary, 7 of those with DCR and good drug toleration after the induction phase terminated treatment due to disease progression (n = 2%), medication toxicity (n = 2, 29% each), or death (n = 3, 43%) (Figure 1). 

Interestingly, most events leading to treatment failure (toxicity or PD) occurred during the first eight months of immunotherapy. Afterward, we observed a plateau of the curve with disease control rate (Figure 2A–C). Moreover, interestingly, those with DCR after the induction phase remain also controlled in the follow-up (Figure 2D). 

Patients who continued immunotherapy at 12-month follow-up had no central nervous system metastasis (n = 0 vs. n = 10, *p* = 0.03) and more frequent used immunosuppressive agents or steroids due to mild-moderate AEs immunotherapy (n = 4 vs. n = 11, *p* = 0.04). Other factors were not important in the subgroup analysis.

### 3.4. Hepatotoxicity, Pneumonitis, and Thyroid Gland Disturbances Were the Most Frequent Adverse Events Related to the Combined Immunotherapy 

As expected, 67% (n = 34) of the patients experienced AEs related to immunotherapy toxicity. Most of them reported one or two AEs (n = 22 [43%] and n = 10 [20%], respectively). Only two (4%) patients had three AEs. The most common AEs was hepatitis (n = 13, 38%), followed by pneumonitis (n = 6, 18%), hypothyroidism (n = 5, 15%), hyperthyroidism (n = 4, 12%), colitis/ diarrhea (n = 4, 12%), rash/vitiligo/pruritus (n = 3, 9%), and hypophysitis (n = 2, 6%) [23]. Other, less frequent AEs (such as lymphangitis hemochromatosis, hematological, sarcoidosis, neurological, diabetes mellitus, and asthenia) were reported in 11 (32%) individuals. The severity of the AEs is presented in Figure 3. Of note, more severe AEs (G3/G4 according to CTCAE v.5.0 (U.S. Department of Health and Human Services) [22]) occurred in half of the all AEs group (n = 17) and were a reason for the therapy withdrawal in 13 of them (77%) after a median time of 3 (range: 2–3) months. However, interestingly, 6 of those 17 patients (47%) had an ORR, and another 4 had SD after the induction phase (DCR 77%). Most AEs occurred during the first six months of observation (Figure 4C), and their risk decreased during the follow-up (HR: 0.41, 95%CI [0.23–0.72], *p* = 0.002). 

Among the 34 patients who experienced AEs, 18% (n = 6) required transfusion of blood and blood products, and 53% (n = 18) needed steroids for a median of 1 (range: 1–2.5) month and other immunosuppressants, including mycophenolate mophetil and cyclosporine used by 3 (9%) and 1 patient (3%), respectively. None of the patients died due to immunotherapy-related AEs.

Interestingly, patients with reported AEs had an 80% lower risk of death (HR 0.2, 95%CI [0.07–0.57], *p* = 0.001) and progression of the disease (HR 0.2, 95%CI [0.09–0.47], *p* < 0.0001) during 12 months of follow-up. In addition, the median OS and PFS for patients who experienced AEs were not reached, while for those without any AEs were 5 (range: 2–15) and 3 (range: 1–3) months for OS and PFS, respectively (Figure 4A,B).

### 3.5. The Combined Immunotherapy Duration and Presence of Adverse Events Were Beneficial Predictors of the 12-Month Follow-Up Outcomes 

Considering the whole patient group, 16 (32%) persons died, and 34 (68%) were alive at the 12-month checkpoint. Of the latter, 14 (41%) continued the immunotherapy, 6 received BRAF/MEK inhibitors (18%), 3 a second-line chemotherapy (9%), and the remaining 11 (32%) subjects were treated only by surgery and/or radiotherapy, or surveillance (Figure 1). The median duration of the immunotherapy was 3 (range: 2–6) months, and the patients received a median number of 4 (range: 3–6) immunotherapy cycles. As expected, in a pairwise comparison, those who were disqualified from immunotherapy had the worst outcome (*p* < 0.01). On the other hand, in log-rank tests, there was no difference between second-line treatment modalities (*p* = 0.65).

The median OS in the whole patient group was not reached (NR, range: 6.8 months-NR). In those who had immunotherapy stopped, the median TTF was 3 (range: 2- NR) months and PFS was 6.3 (range: 3-NR) months. Interestingly, patients with SD had similar outcomes to those with CR + PR in 12-month follow-up (*p* = 0.69). The HR for death in patients with CR + PR vs. PD was 0.17 (95%CI [0.04–0.79], *p* = 0.023). On the other hand, HR for death in patients with SD vs. PD was 0.46 (95%CI [0.13–1.65], *p* = 0.23), (Figure 3D). 

In a single Cox regression model, patients with central nervous system metastases had a four times higher risk of death (HR 4.1, 95%CI [1.4–12], *p* = 0.005), similarly to those with performance status 1 (HR 3.2, 95%CI [1–3.9], *p* = 0.0036) and requiring transfusion of blood and blood products during treatment (HR 3.4, 95%CI [1.1–11), *p* = 0.023). Interestingly, immunosuppression during treatment did not influence the outcome but was even a protective factor (HR 0.11, 95%CI [0.015–0.87], *p* = 0.036, HR 0.18, 95%CI [0.039–0.78], *p* = 0.02, for immunosuppressants and steroid use, respectively). Similarly, patients who received immunotherapy in the adjuvant setting had a better outcome and more prolonged survival than those not exposed to such treatment in the past (HR 0.53, 95%CI [0.28–0.99], *p* < 0.001).

In a multiple Cox regression model, the number of applied immunotherapy series (HR 0.6, 95%CI [0.47–0.85], *p* = 0.002) and the presence of AEs during treatment (HR 0.27, 95%CI [0.12–0.67), *p* = 0.005) were independent predictors of better PFS, while metastasis to the central nervous system (HR 2.8, 95%CI [1.11–7.08], *p* = 0.029) influenced it negatively (*p* <0.0001 for the model).

## 4. Discussion

In this study, we have documented 12-month treatment outcomes using combined immunotherapy in metastatic melanoma patients in real life in Poland. Our results showed that early clinical results after the induction phase are crucial for further prognosis. Most patients with disease control in the first CT scan continued the treatment till the 12-month checkpoint. Furthermore, even if treatment was withdrawn due to toxicity, it may lead to disease control, as in 70% of our patients.

However, comparing the results of the pivotal studies [25], the outcome for polish patients is worse, especially in terms of early treatment results represented by PFS values. In the Checkmate 067 trial, after 60 months of follow-up, the median OS was NR and the median PFS was 11.5 months (95%CI [8.7–19.3]) with 58% of ORR with most progressions during the first 3 months of treatment. In comparison, in our study the median OS was NR whereas PFS was 6.3 months, with ORR in 34% of the patients. Other real-life studies are very limited in the literature. In the study from Israel, ORR was reached 61% with a median PFS of 12.2 months and a median OS not reached after 12 months of follow-up [26]. In the USA report comparing different front-line treatment modalities, the median OS for the immunotherapy combination was not reached, while median OS for patients treated with front-line anti-PD-1 or BRAF/MEK inhibitors was 39.5 and 13.2 months, respectively [27]. There are several possible explanations for these differences. First, that treatment modality is available in Poland for a short time, so we are still on the learning curve in toxicity management and proper patient qualifications. On the other hand, the population in which this treatment was applied is significantly different from the precisely selected individuals in the pivotal clinical trial. Taking into account known factors of bad prognosis, in our study, compared to patients in the Checkmate 067 trial [25], the number of patients with PS 0 was lower (44% vs. 75%), and more patients had increased baseline LDH levels (48% vs. 35%), brain metastases (20% vs. 2.5%) and BRAF mutation (40% vs. 31.6%), which represent unfavorable risk factors. Interestingly, in Asia, combined immunotherapy is considered similarly effective to a single immunotherapy in advanced melanoma patients [28]. However, the acral and mucosal melanoma rate is higher in that population. These subtypes likely had a lower mutation burden; therefore, the response to immunotherapy is considerably worse [29,30]. 

Importantly, our study demonstrated that immunotherapy-related AEs might be beneficial, even if the patients need immunosuppressive treatment. This finding is supported by other studies that also indicated the protective role of AEs [31,32,33,34]. For example, it has been shown that skin rash and vitiligo may be associated with OS benefits (HR 0.45, 95%CI [0.251–0.766] and HR 0.22, 95%CI [0.025–0.806], respectively) in melanoma patients treated with nivolumab [31]. Furthermore, Keller et al. [31] reported better results in individuals who experienced 3 or more episodes of AEs, whereas in Asher et al. [26], patients without AEs survived only 1.3 months. Similarly, as in our study, most AEs in literature data occur in the combination phase rather than in nivolumab maintenance [35].

The occurrence of AEs is related to the activation of the T cell response. In contrast, impaired function or insufficient T cell generation suggests resistance to immunotherapy; therefore, these events are mutually exclusive [36,37]. Noteworthy, combined immunotherapy is associated with the four times higher frequency of adverse events than chemotherapy or anti-PD-1 monotherapy and 2.5 times higher than BRAF/MEK inhibitors [38]. In another real-life study from Poland [39], which aimed to evaluate anti-PD-1 monotherapy, the rate of G3/G4 AEs was 6% which was slightly lower than in reports from other countries where it ranged between 8.7% and 15% [40,41,42] comparing to 50% in combined immunotherapy as we reported The protective effect of immunosuppressants is possibly explained by the strong correlation between their use and the presence of AEs. Some studies also demonstrated a favorable outcome in patients who received systemic steroids [31]. However, in the Keynote 054 trial with adjuvant pembrolizumab, although the presence of AEs was associated with a longer RFS, the effect of the treatment was lower after the use of systemic steroids for AEs (HR 0.5, 95%CI [0.23–1.07]) versus without steroid use (HR 0.34, 95%CI [0.21–0.56]) [32]. However, our results suggest that in the case of severe AEs, it is better to use steroids and/or immunosuppressants for the final outcome than discontinue immunotherapy. That is likely an essential remark, which should be taken into account by oncologists. The high discontinuation rate due to AEs (almost 40% in our study) and too early treatment withdrawal (38% of patients received less than 4 series) despite the lack for PD seems to be the wrong strategy. Prompt AEs management and, if possible, immunotherapy continuation are important since they might improve the patient’s prognosis as we demonstrated in the case report from our clinic [43]. For example, a study from Nova Scotia demonstrated that only half of immunotherapy-related AEs are managed according to guidelines and about 17% of discontinued patients were withdrawn too early for this reason [44]. The authors postulated that it may be caused by the aversion of some physicians to administer high-dose steroids, as it potentially may inhibit the immunotherapy effect. However, there are not sufficient data to support this hypothesis as we mentioned before. 

The distribution and frequency of AEs vary between immune monotherapy and its combination. In our study, the frequency of AEs was slightly different than in other reports, where diarrhea is the most common event (45% vs. 12% in our study), followed by pruritus (35%), rash (30% vs. 9% for skin AEs), and hepatitis (14–18% vs. 38%) [36]. Grade 3 and 4 AEs in patients treated with the combination of ipilimumab 3 mg/kg and nivolumab 1 mg/kg in the clinical trial were reported in 55% of patients [36], but occurred only in 34% in our study. 

The main limitation of our study is the small number of patients but the study was powered enough (88%) to show the efficacy in therapy used. The other limitation is relatively short observation time, which is related to the reimbursement policy in Poland. Nevertheless, it is noteworthy that real-life data on combined immunotherapy in advanced melanoma are very scarce in the literature, and each report is valuable in that field. Furthermore, we retrospectively analyzed patient records. However, our project has a prospective format and we plan to follow up on the enrolled individuals in the future. 

## 5. Conclusions

Demonstrated outcomes are considerably worse in terms of PFS than those in the pivotal trial and other real-life studies. For the patient’s prognosis, critical are the first weeks of immunotherapy. After that, early responders are likely to continue the treatment successfully. 

The appearance of AEs is a beneficial predictor; thus, proper AEs diagnosis and management and immunotherapy continuation are critical for the outcomes. In severe AEs, despite immunotherapy withdrawal, the disease may remain under control. Most AEs occur during the induction phase; thus, patients must be closely monitored during immunotherapy’s first months. 

These findings may be valuable for the future patients management and cooperation between melanoma-oriented centers in Poland. Apparently, there is a lot to improve but, as mentioned, we are still gaining experience, and we hope that our next follow-up will be more favorable.

## Figures and Tables

**Figure 1 biomedicines-10-02528-f001:**
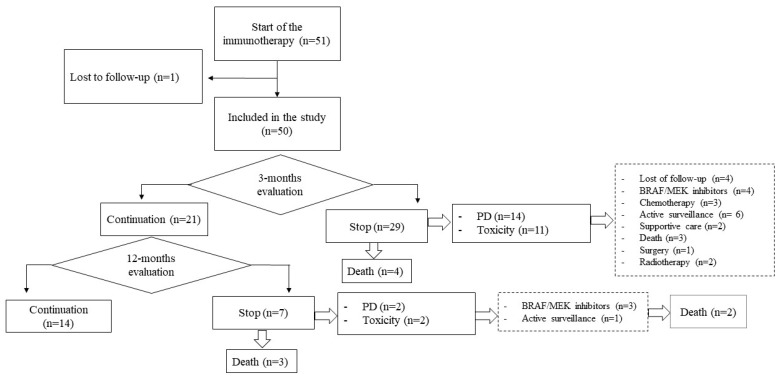
Profile of enrolled patients.

**Figure 2 biomedicines-10-02528-f002:**
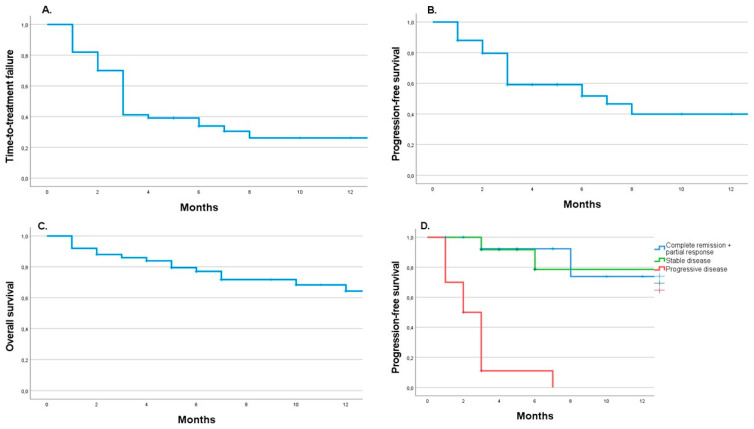
Kaplan-Meyer curves for time-to-treatment-failure (**A**), progression-free survival (**B**), and overall survival (**C**) in the whole group of patients and progression-free survival depending on the results of first imaging (**D**).

**Figure 3 biomedicines-10-02528-f003:**
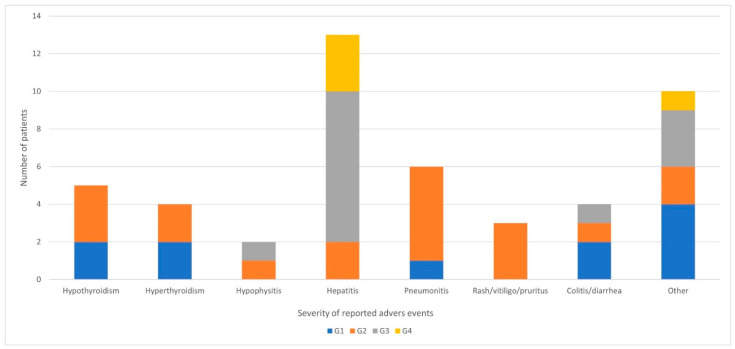
Prevalence and grade (according to CTCAE version 5.0, U.S. Department of Health and Human Services) [22]) of immunotherapy-related adverse events.

**Figure 4 biomedicines-10-02528-f004:**
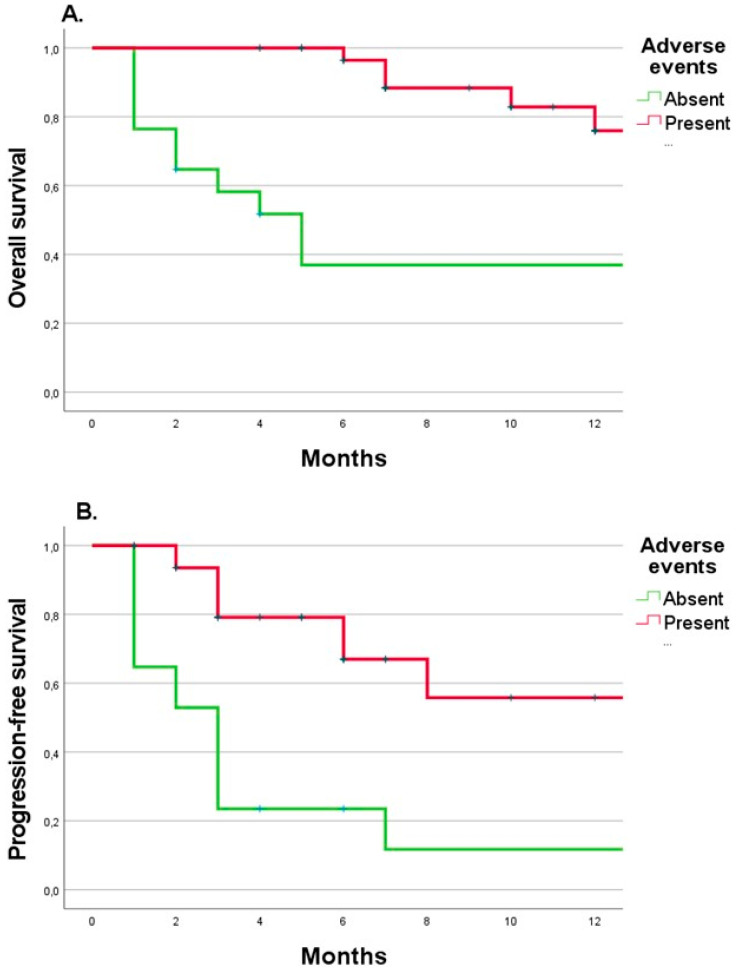
Kaplan-Meyer curves for overall survival (**A**) and progression-free survival (**B**), depending on the presence of adverse events and the time to occurrence of the first adverse event (**C**).

**Table 1 biomedicines-10-02528-t001:** Demographic and clinical characteristics of melanoma patients at enrollment (n = 50), including past oncological treatment and basic laboratory test results.

Patients Characteristics, n = 50		
Demographics		
Age		57.5 (46–65)
Males, n(%)		26 (52)
Body-mass index (kg/m^2^)		27.9 (23.5–30.1)
Comorbidities		
Hypertension, n(%)		18 (36)
Ischemic heart disease, n(%)		2 (4)
Autoimmunologic diseases, n(%)		2 (4)
Hypothyroidism, n(%)		2 (4)
Diabetes mellitus type 2, n(%)		5 (10)
Hypercholesterolemia n(%)		9 (18)
Baseline melanoma-specific characteristics		
Performance status, n(%)	0	22 (44)
	1	28 (56)
TNM stage IV AJCC 8th edition, n(%)	M1a	6 (12)
	M1b	3 (6)
	M1c	31 (62)
	M1d	10 (20)
Number of disease sites, n(%)	≤3	30 (60)
	>3	19 (38)
	Unknown	1 (2)
Site of metastasis at enrollment, n(%)	Lymph nodes	34 (68)
	Subcutaneous tissue	17 (34)
	In-transit	7 (14)
	Liver	19 (38)
	Central nervous system	10 (20)
	Lungs	25 (50)
	Bones	13 (26)
	Other	19 (38)
BRAF mutation, n(%)	Wild-type	30 (60)
	V600E	16 (32)
	V600K	3 (6)
	Undetermined	1 (2)
PD-L1 expression > 5%, n(%)	Present	4 (8)
	Absent	28 (56)
	Not available	18 (36)
Basic laboratory parameters		
Hemoglobin (g/dL)		13.1 (11.2–14.6)
Neutrophils (10^3^/uL)		4.8 (3.6–6.2)
Lymphocytes (10^3^/uL)		2.1 (1.6–5.3)
Platelet count (10^3^/uL)		280 (210–337)
Lactate dehydrogenase (U/L)		209.7 (175.4–472)
Estimated glomerular filtration rate (mL/min/1.73 m^2^)		76 (60–90)
Alanine transaminase (U/L)		21.9 (11.8–33.6)
Aspartate aminotransferase (U/L)		20.5 (16–30.2)

Categorical variables are presented as numbers (percentages), and continuous as median and interquartile ranges. Abbreviations: BRAF—B-Raf proto-oncogene, PD-L1—programmed cell death ligand 1, TNM 8th AJCC—8th edition of Tumor Node Metastasis staging system of melanoma according to American Joint Committee on Cancer.

## Data Availability

Not applicable.

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
