# Peer review of "Early Effects of Nivolumab and Ipilimumab Combined Immunotherapy in the Treatment of Metastatic Melanoma in Poland: A Multicenter Experience"

_biomedicines, 2022, doi:10.3390/biomedicines10102528_

Round 1

Reviewer 1 Report

Dear Authors,

I have read with interest the manuscript and I think that it is very intersting and well written, even if the number of patients is low (50) and it could be ralated with  the type of disease.

However I send you my comments:

1) Please add the power calculation

2) Please add a group of patients treated with conventional drugs in order to evaluate the difference in the development of ADRs 

Author Response

I have read with interest the manuscript and I think that it is very intersting and well written, even if the number of patients is low (50) and it could be ralated with  the type of disease.

Answer: Thank you for your appreciation of our study.

Please add the power calculation

Answer: The power for Cox regression is 82%, and the general power of the study is 88%. The information has been added to the manuscript.

Please add a group of patients treated with conventional drugs in order to evaluate the difference in the development of ADRs 

Answer: Our study is a retrospective, real-life report without a control group and aimed to evaluate the efficacy of combined immunotherapy in metastatic/advanced melanoma. Due to the heterogeneity of the population treated with other agents (e.g., anti-PD-1 monotherapy/targeted therapy/ chemotherapy) in terms of adverse prognostic factors such as disease dynamics, disease burden, presence of brain metastases, level of LDH, etc. (ESMO Clinical Practice Guidelines, 2019) it would be very difficult or even impossible to appropriately select patients in subgroups to avoid major differences between them. It would also provide ethical concerns since combined immunotherapy is currently a standard of care for metastatic melanoma patients. However, considering data from clinical trials in the Checkmate 067 study (Wolchok et al. 2017) where nivolumab was a control arm for combined immunotherapy the frequency of G3-G4 adverse events was 24%  vs. 59%. On the other hand, in the Cochrane Metaanalysis (Pasquali et al. 2017) combined immunotherapy had increased toxicity in an indirect comparison with chemotherapy (RR 3.49, 95% CI 2.12 to 5.77),  BRAF/MEK inhibitors (RR 2.50, 95% CI 1.20 to 5.20) as well as anti-PD-1 monotherapy  (RR 3.83, 95% CI 2.59 to 5.68). On this basis, we may speculate upfront that adverse events would be less frequent in other treatment modalities. However, we appreciate your comment and think it is valuable the readers know the data regarding data for single agent immunotherapy also in real-life context. We extended our manuscript with some data on that issue. We compared differences in ADRs referring to Checkmate 067 study in the Introduction section, and we included information about real-life data in the Discussion section.

Reviewer 2 Report

This study is very well done. Provides comprehensive information on the effect of combined immunotherapy in Poland. In the context of the original clinical multicenter study, the work is important in the comparison of individual data on the effect and, above all, on adverse effects. These are the main reason for early termination of therapy.

In the introduction - it is not necessary to write (indicate) malignant melanoma - just melanoma is enough... it is a tumor that does not have a benign variant, every melanoma is malignant, the correct designation based on the pathological anatomical nomenclature is therefore only melanoma.

Author Response

This study is very well done. Provides comprehensive information on the effect of combined immunotherapy in Poland. In the context of the original clinical multicenter study, the work is important in the comparison of individual data on the effect and, above all, on adverse effects. These are the main reason for early termination of therapy.

Answer: Thank you for appreciation of our study.

In the introduction - it is not necessary to write (indicate) malignant melanoma - just melanoma is enough... it is a tumor that does not have a benign variant, every melanoma is malignant, the correct designation based on the pathological anatomical nomenclature is therefore only melanoma.

Answer: The text has been corrected accordingly.

Round 2

Reviewer 1 Report

Dear authors I have read the revised version and I think that the manuscript has been improved.

I have not other comments